# The Effect of Decision Training, from a Cognitive Perspective, on Decision-Making in Volleyball: A Systematic Review and Meta-Analysis

**DOI:** 10.3390/ijerph17103628

**Published:** 2020-05-21

**Authors:** Manuel Conejero Suárez, Antonio Luiz Prado Serenini, Carmen Fernández-Echeverría, Daniel Collado-Mateo, M. Perla Moreno Arroyo

**Affiliations:** 1Faculty of Sports Sciences, University of Extremadura, 10003 Cáceres, Spain; mconejeros@unex.es; 2Federal Center for Technological Education of Minas Gerais, Varginha 37022-560, Brazil; alpserenini@yahoo.com.br; 3Faculty of Education Sciences, University of Sevilla, 41013 Sevilla, Spain; 4Centre for Sport Studies, Rey Juan Carlos University, 28943 Madrid, Spain; daniel.collado@urjc.es; 5Faculty of Sports Sciences, University of Granada, 18071 Granada, Spain; perlamoreno@ugr.es

**Keywords:** perceptual training, cognitive training, decision-making, volleyball

## Abstract

Over the past few decades there has been great interest in the study of cognitive processes, and specifically decision-making, from a cognitive perspective. The aim of the present study was to systematically review the scientific literature on the effect of decision training interventions/programs, from a cognitive perspective, on the decision-making of volleyball players. The systematic search was carried out in five scientific electronic databases according to PRISMA guidelines Web of Science (WOS), Pubmed (Medline), Scopus, SportDiscus and Google Scholar. A total of eight studies met the inclusion criteria. The main finding of the meta-analysis was that the use of decision-making training programs/interventions led to significant improvements in volleyball players’ decision-making (Standardized mean difference = 0.94 with 95% confidence interval from 0.63 to 1.25), compared to normal active volleyball training. In addition, the heterogeneity of the interventions was low (I^2^ = 0%). From the results of the studies analyzed, we recommend using decisional interventions or training, both as part of normal active training or complementary to it, to improve the decision-making of the players, thus optimizing their ability to perceive and process relevant stimuli, and then generate quick and effective responses. These findings can be useful in the process of sports training.

## 1. Introduction

The study of cognitive processes, and specifically decision-making, has been of great interest to researchers in recent years [1]. Decision-making in sport is highly relevant because it is essential for the achievement of sporting expertise [2,3].

According to Raab [4] “decision-making can be defined as the process through which athletes choose a technique appropriate for their current game situation considering their context “(p. 4). This process is complex, as it depends on athletes’ abilities to detect the right information in the environment, to plan future actions, and select the most appropriate response based on the specific situation of play [5], although successful decision-making can result in achieving the ultimate goal of a given action [6].

Obtaining a detailed understanding of the decision-making process has been considered an important aim in research [7,8,9]. As such, numerous studies have analyzed the various processes involved when athletes try to select the correct response in sport situations [10]: anticipation [11,12], attention [13,14], experience [15,16], decision-making [17,18], memory [19,20], mental images [21,22], and perception [23,24].

Because of this, studies in sport have addressed decision-making from different perspectives. Most have tried to understand and explain the decision-making process in sport to improve performance [6]. The two fundamental perspectives for the study of these decision-making processes are: the ecological perspective [25] and the cognitive perspective [26].

The ecological perspective according to Araújo et al. [27] defines “sport as a dynamic system of constant interactions between the subject and the environment” (p. 8). During the interaction between an individual and their environment, including other individuals, athletes learn to perceive information, leading to the development of perceptual mechanisms that help capture the most important stimuli within the game context [28]. According to this perspective, during the decision-making process, the athlete collects information specific to the environment, perceiving the significant properties of the environment without mediating processes or interpretation of the information, and then issues a response [29]. Therefore, subjects in sporting situations receive information from the environment and act via a mechanism of perception-action: there is no need for the intervention of mental representations [30].

Conversely, the cognitive perspective purports that decision-making occurs prior to action, and is done based on perceptual processing that occurs prior to the processing of information [31]. From this perspective, in sports situations, athletes analyze the environment where the action unfolds to obtain the most relevant information, interpret this information using mental representations and cognitive processes, and then select an appropriate response [32]. From a cognitive perspective, the mechanism of information processing is highlighted. This mechanism is based on the athlete’s cognitive strategies that occur in working memory, knowledge structures, and anticipation processes [17].

Within the cognitive perspective, two approaches have been posited for studying and understanding decision-making: the first focused on perceptual mechanisms (through the study of visual and temporal parameters) [31], and the second focused on memory-related processes [33].

Athletes’ perceptual skills are hugely important in the study of decision-making considering visual and temporal parameters. According to this approach, the subject must quickly perceive and interpret information in the environment to have enough time to plan, initiate, and execute the sport skill [34]. Moreover, visual search strategies allow the athlete to extract relevant information from the environment, thus favoring an anticipated response [35]. Similarly, the approach includes temporal parameters, which are defined as the time between two processes: the selection of the stimulus (perceptive component) and the selection of the response (cognitive component). This reaction time will be influenced by previously perceived stimuli [36].

On the other hand, the performance of athletes will depend on the mental representations and cognitive processes that must be carried out between the interpretation of the stimulus and the selection of the response [37]. Therefore, the athlete’s knowledge about the sport will form a basis to favor the selection of the correct answer [38,39].

It is important to pursue the improvement of decision-making given its essential role in performance [40]. The study of decision-making has demonstrated that perceptual-cognitive processes, which influence it, are trainable [41]. For this reason, research on decision-making has focused on developing and implementing interventions/training programs that improve decision-making in athletes [42].

Various programs and strategies based on the cognitive model with visual and temporal parameters have been used in decision-making training to improve athletes’ abilities to detect information from the environment [43]. These training programs aim to improve athletes’ understanding of information and focus on effective visual search strategies [43]. This causes the athlete to develop a series of skills that make it easier to recognize and remember different playing patterns, discriminating against irrelevant stimuli, and thus improve anticipation, decision-making, and action outcome [43]. Some of these training programs have included: viewing and simulation of game sequences [44], temporal and spatial occlusion [45], occlusion of the action sequence and feedback on accuracy [46], or manipulation of attention orientation through visual video signals [47]. 

Decision-making training based on memory-related processes is based on the need to provide athletes with experiences that help them be thoughtful, autonomous and able to make their own decisions [40]. In this type of training, various techniques are used, including player questioning or video feedback [4], the presentation of images or videos during training or competition to analyze tactical behavior [48], the use of mental imagery [49], and feedback [50].

Due to the need to know the effectiveness of such programs for improving decision-making in open skill sports such as volleyball, it is necessary to carry out a systematic review and a meta-analysis on this topic of research. Volleyball is a collaboration-opposition sport, with mandatory rotation of the players, in which each team is in separated courts without the possibility of invading the opposite field. Those characteristics mean that volleyball involves frequent exchanges in the ball possession and players must respond to them by making decisions with and without the ball [51]. Given that players are not allowed to catch the ball, there is a temporary deficit in the different game actions in volleyball [52], which hinders the decision-making process [53]. 

Considering past findings [49,54,55], we hypothesized that decision-making training programs/interventions based on perceptual mechanisms and memory-related processes would improve the decision-making of volleyball players.

Appropriate analyses should be carried out in order to know if the hypothesis is fulfilled, and to test the magnitude of the observed effects, if any. At present, there has yet to be a meta-analysis of studies that pursue the improvement of decision-making in volleyball through intervention programs based on a cognitive perspective. The current systematic review and meta-analysis aimed to review randomized and non-randomized controlled trials to evaluate whether decision training from the cognitive approach is more effective than the typical training to improve decision making in volleyball players.

## 2. Methods 

To perform the current review, we adopted procedures from previous review and meta-analysis studies [56,57]. Further, we considered reporting standards and guidelines from systematic review and meta-analysis protocols (PRISMA) [58].

### 2.1. Inclusion and Exclusion Criteria

In order to select the manuscripts included in the present study, this approach was followed: (1) the studies were based on interventions or training programs for decision-making from the cognitive perspective; (2) aspects of player decision-making were evaluated; (3) all articles pertained to volleyball; (4) English or Spanish versions of the manuscripts; (5) Articles were posted in the present century; and (6) articles were original research. 

On the other hand, the following exclusion criteria were set: (1) studies from the ecological approach; (2) studies on beach volley; (3) studies focused on the improvement of declarative of procedural knowledge; (4) designs without a control group.

### 2.2. Search Strategy

The systematic literature search was carried out using the PRISMA guidelines [58] in Web of Science (WOS), PubMed (Medline), Scopus, SportDiscus, and Google Scholar. The search was performed considering articles published in the last 10 years. The following syntax, in two different languages (English and Spanish) was used for the search process: (“questioning” or “video-feedback” or “image viewing” or “visual search strategies” or “reflective monitoring”) and (“reaction time” or “response time” or “visual function” or “anticipation” or “spatial parameters” or “temporal parameters”) and (“volleyball”) and (“cognitive model” or “cognitive perspective” or “perceptual mechanism” or “cognitive training” or “decision-making training” or “perceptual training”) and (“decision-making”) and (“intervention” or “experimental” or “quasi-experimental” or “experimental group” or “control group”).

### 2.3. Assessment of Risk of Bias

For the assessment of risk-of-bias, we used the Evidence Project risk of bias tool, which is a simple and reliable tool to evaluate the study design (items 1–3), the bias that may affect the equivalence of the groups or the external validity of the results (items 4–6) and the potential bias from between-group differences at baseline (items 7–8) [59]. The main advantage of this tool is the applicability in both randomized and non-randomized controlled trials. 

### 2.4. Study Selection and Data Collection

The study selection was conducted following the PRISMA guidelines. First, two of the authors (M.C.S. and C.F.-E) reviewed and manually removed duplicated articles. After that, those articles that did not fulfill the criteria were excluded. In case of disagreement between these two authors, a third author (M.P.M.A) was consulted. 

Secondly, two of the authors from this manuscript extracted data from the articles included in the meta-analysis. This information was then collected and verified by a third author. The information was extracted and reported following the PICOS approach: participants, age, level of play, country and the study design (PICOS) [60]. 

### 2.5. Statistical Analysis

The Review Manager Software (RevMan, 5.3) was used for data analysis [61]. The standardized mean differences (SMD) was calculated given that different tools and units of measurement were reported in the included studies. The inverse variance test was utilized to generate SMD, which was interpreted according to the Cochrane Handbook of Systematic review as “small” when SMD < 0.4, “moderate” for values between 0.4 and 0.7, and “large” when SMD was higher than 0.7 [62]. When the outcomes were assessed using scales with opposite directions (as happened in the article by Fleddermann et al. [54] and Formenti et al. [55]), the less common was multiplied by −1 [63]. The choice between random or fixed effects models was made according to the level of inconsistency, considering the cut-off point at I^2^ > 40% [64]. Given the very low inconsistence observed in the current meta-analysis (I^2^ = 0%), a fixed effects model was chosen.

## 3. Results

### 3.1. Study Selection

Figure 1 (PRISMA flow diagram) shows the process that has been carried out during the systematic review for the selection of the different studies. The initial search using the syntax detailed above identified 32 articles in total from the following electronic databases: WOS (7), PubMed (1), Scopus (8), SportDiscus (5) and Google Scholar (11). Three articles were eliminated because they were duplicates. Of the remaining 29 articles, 15 were eliminated because they were not related to the subject of the study, two were eliminated because the decision-making training programs/interventions were based on the ecological perspective, two were eliminated because they measured tactical knowledge and not decision-making, and two because they did not have a control group. Based on these exclusions, eight studies were included in the meta-analysis, each testing the effect of an intervention/training program based on the cognitive perspective on decision-making (Figure 1). 

### 3.2. Study Characteristics

Data were extracted following the PICOS approach. In this regard, Table 1 summarizes the main characteristics of the participants (P) and the study design (S), while Table 2 shows the characteristics of the intervention (I), and the comparison (C), as well as the outcome measure (O). 

In the different studies, there were a total of 243 participants. Of these, 97 were distributed in the experimental group, 94 in the control group, and 52 in other groups that were not included in the analysis. Six studies were conducted in youth categories from regional leagues of different countries (Italy, Spain, Greece and Algeria), one in a Brazilian State Volleyball Championship, and one with players from the 1st, 2nd and 3rd German divisions.

Next, we summarize the following details of the decision-making training programs/interventions: duration, number of training sessions and type of decision-making training programs/interventions. In the study by Fleddermann et al. [54] the players received an 8-week training program of two workouts per week. In each session, the players had a 30-min intervention, in addition to their usual training, divided into three phases of 8 min, with a rest of 3 min between each. In this intervention, players performed perceptual-cognitive tasks via 3D-MOT, with motor tasks that were either specific (blocks, sets, attacks) or non-specific (perform jumping) to volleyball.

In the study by Formenti et al. [55] the training program lasted 8 weeks with each training session lasting 80 min. The sessions were divided into warm-up (10 min), perceptual intervention program (30 min), volleyball exercises (20–30 min), and cool down (10 min). The intervention program was divided into different stages of visual search tasks, with each task lasting 6 min. 

In the study by Fortes et al. [49] the intervention program consisted of 8 weeks with a total of 24 sessions, each separated by 48 h. These sessions were held 30 min after each physical/technical training session and lasted 10 min. Players from two groups (control and experimental) participated in the same physical/technical training sessions. During the intervention, the experimental group undertook training based on the observation of images and videos of successful volleyball actions in competitive events. This training was designed to facilitate imaginative capacity. In addition, to generate emotions athletes were asked to consider an imaginary situation, in the first person, that would be close to the reality of a competition situation. During the same training sessions, the control group viewed videos related to sportswear ads (caps, T-shirts and shorts).

Gil-Arias et al. [65] involved an 11-week intervention in which a program was applied during the training sessions. The workouts lasted 120 min and were divided into two phases: firstly a 60-min technical-tactical phase, and a second phase, where the intervention program was implemented, with a 6 vs. 6 game situation lasting 60 min. During the study, all 8 players trained at the same time and competed equally, although only those in the experimental group were subjected to the program, which required players to analyze their own decisions. Specifically, players viewed videos of their performances and gave comments (video-feedback), helped by questioning. Decision-making in the attack actions performed by both groups in all competition matches during the intervention period (11 matches) were assessed. 

In the study by Lola et al. [66], players experienced a four-week intervention designed to improve decision-making related to the serve in volleyball. The intervention involved a program applied 3 times a week (12 practice sessions). Each session was 70 min long with the first 30 min dedicated to training by watching volleyball videos followed by 10 min of warm-up and 30 min practicing the action considered in the video. Specifically, each player performed 20 serves to a team with six opponents, looking for free zones in the opponent’s court.

Merzoug et al. [67] conducted a training program for improving various aspects of decision-making such as speed and decision accuracy. The study evaluated basic volleyball situations through simulation. The program was based on tracking multiple objects during four 3D game actions (serve, setting, blocking and defense). The program consisted of perceptual learning situations through occlusion training, play analysis and tactical volleyball discussions, as well as feedback from coaches.

In the study by Moreno et al. [40], the authors conducted an intervention comprising 13 competition matches. The program involved holding various supervisory meetings with each of the players of the experimental group between 24 and 48 h after each match. These meetings analyzed the attack situations performed by the players during these matches. During the meetings, players watched videos of their performances during the match and then offered their own analyses (video-feedback). The supervisor/mentor also asked questions (questioning) to help develop the players’ reflective abilities.

Finally, Sáez-Gallego et al. [43] applied an eight-session training program over the period of one month (two weekly sessions of 20 min each). The goal was for players to use an effective visual pattern that would allow them to take advantage of highly informative areas at the time of the action. The training session consisted of four parts: (1) an attention orientation video with 16 setting sequences (eight slowed and eight normal speed); (2) training with feedback on the set direction (24 frozen sequences at the time of decision); (3) training with feedback on reaction time (24 sequences, where 12 were cut at the key time and 12 edited with light signal); and (4) random practice with 12 setting sequences without changing speed or duration.

### 3.3. Risk of Bias

Table 3 showed that there was a potential risk of bias due to randomization of assignment and selection. In this regard, five of the eight studies did not conduct a random assignment of participants to the intervention. As expected, there was not a random selection of participants due to the characteristics of the target population. Another potential risk of bias was the differences at baseline in two studies, which could influence the results.

### 3.4. Interventions

The decision-making training programs/interventions involved video feedback and questioning, training based on image or video reproduction, three-dimensional training with multiple objects, training through visual search tasks, or training using computer simulations. The duration of the interventions varied between 4 and 13 weeks. The total number of sessions within the programs raged from 8 to 26 sessions, and the durations of these interventions/programs ranged from 10 to 60 min. The studies carried out by Formenti et al. [55] and Gil-Arias et al. [65] did not specify the number of sessions carried out in the program, but the duration of the program (8 and 11 weeks, respectively). The study by Merzoug et al. [67] did not specify either the duration of the program or the number of sessions carried out.

### 3.5. Outcome Measures

Figure 2 shows the effects of decision-making training programs/interventions, based on cognitive perspective, on the decision-making of volleyball players. To evaluate the decision-making of athletes in three of the articles, the Game Performance Assessment Instrument (GPAI) elaborated by Oslin, Mitchell, y Griffin [68]. Two other investigations used Superlab. The remaining three investigations each used a different instrument, these being: the program NeuroTracker^TM^ Core de CogniSens Athletics Inc. (Montreal, Canada) the reaction time test, the court performance test, and anticipation test.

The meta-analysis revealed that the interventions/training programs, based on the cognitive perspective, significantly improved the decision-making of athletes in the experimental groups compared to those in the control groups in all studies (see Figure 2). These improvements would mean that players are able to better identify and process the relevant stimulus and make a more effective decision based on the perceived information. The average effect measured through the SMD was 0.94, with a 95% CI from 0.63 to 1.25. Following the proposed classification, the size of this effect was large. The level of heterogeneity was low (I^2^ = 0%). Since two studies could be biased by differences at baseline [43,67], we also conducted the meta-analysis excluding them to test the potential influence of that bias. The results indicated a SMD of 0.91 with a 95% CI from 0.57 to 1.24 and I^2^ = 7%. Thus, the bias caused by those differences at baseline did not influence the results of the meta-analysis.

## 4. Discussion

The present study aimed to systematically review the scientific literature on the effect of decision-making training programs/interventions, based on cognitive perspective, on the decision-making of volleyball players. The result of the analysis showed that training programs/interventions based on cognitive perspective led to a significant improvement in the decision-making of athletes in the experimental groups compared to those who only experienced normal active volleyball training. This significant improvement was observed in the eight studies analyzed and can be considered as a large difference based on the size of the effect (SMD of 0.94, with a 95% CI from 0.63 to 1.24 and *p*-value < 0.01). As such, the researchers of this study consider that the application of cognitive training programs based on perceptual training, or those that encourage athlete reflection (which can be used as part of usual active training or in addition to it), represent a benefit to decision-making development.

Of the articles considered in this study, four were focused on improving decision-making through memory-related processes [40,49,65,66]. These studies focused on improving decision-making through video and image viewing and by a supervisor applying questioning and video feedback. This type of training programs improved players’ abilities to analyze technical-tactical actions, thus making it possible for them to make the best decision with greater efficiency [50]. Four other studies were focused on improving decision-making through visual and temporal parameters [43,54,55,67]. These studies focused on improving decision-making via perceptual and simulation training aimed at improving visual search strategies. According to Kenny and Gregory [69], this type of program helps players improve their recognition of environmental signals (e.g., by analyzing and selecting the most relevant stimulus of the opponent tactical reception system during the service [66,67]), allowing a reduction in reaction time (e.g., optimizing the interpretation of the opponent setter movements, enabling the anticipation and improving the effectiveness of the block [43]) and a better success when making decisions (e.g., identifying the optimal trajectory of the ball based on the position of the opponents and the characteristics of the opponent block [40,65]).

The durations of the decision-making training programs/interventions varied between 4 and 13 weeks, with between 8 and 26 sessions. The durations of the programs within each session ranged from 10 to 60 min. These characteristics suggest that decision-making training programs/interventions should last at least 4 weeks, with 8 training sessions, to achieve significant improvements in decision-making. Prior studies focused on improving cognitive processes via memory-related processes in youth categories have recommended that intervention programs last at least 12 sessions [70]. This is because interventions need to be sufficiently extensive to generate significant improvements in decision-making and long-term memory changes [5]. 

Significant improvements in decision-making due to targeted training programs are unrelated to the age or level of the participants: In all studies, regardless of age or level, significant improvements were achieved. However, it remains necessary to consider the age and level of the participants when determining the approach of the programs. Research in youth categories showed that athletes with a higher level of skill than other in the same category of play tended to have faster and more effective decision-making, which favored faster learning and adaptation [71]. Moreover, studies using the expert-novice paradigm show that experts have more knowledge of the sport and this allows them to recognize game patterns, detect relevant information and solve problems more effectively [72]. This makes achieving meaningful improvements in decision-making more complex [73] and, at the same time, more relevant [74]. However, to our knowledge, there are no studies aimed to improve the decision-making processes using decision training from a cognitive perspective in elite or amateur adult volleyball players who meet the inclusion criteria. Therefore, the results of the current systematic review and meta-analysis are limited to young athletes.

Volleyball is an open-skill sport, so the ability to make decisions is an essential component of achieving performance in the different game actions [75]. This type of sport has a complex nature, with athletes constantly making decisions in a highly dynamic and unpredictable setting [76]. This means that athletes must attend to a large number of stimuli, which they will have to perceive and then process to make a decision [77].

A review of the existing literature indicates that this is the first systematic review and meta-analysis aimed at analyzing the effects of decision-making training programs/interventions, based on the cognitive perspective, on the decision-making of volleyball players, using a valid and widely accepted methodology (PRISMA). Although the results are relevant and support the use of decision-making training programs/interventions, more interventions studies are necessary to increase the quality of evidence. Moreover, research indicates that both memory-focused programs [40,49,65,66], and those related to perceptual mechanisms [43,54,55,67] can be useful training tools to improve the perception of stimulus and the selection of the needed action, improving the decision making in sports [78]. Thus, the effectiveness of these interventions programs is largely because athletes anticipate the different actions, being able to select the best response within long-term memory, achieving optimal results in game objectives [50].

Future research should aim to compare decision-making training programs from different perspectives (cognitive and ecological) [79,80]. The present study has limitations that must be taken into account. First, three of the studies used the GPAI instrument to assess decision-making, while the other studies were conducted with other instruments. Secondly, the literature review was conducted in only two languages, Spanish and English, meaning there was a high risk that we excluded other relevant articles written in other languages. In third place, the protocol of the current systematic review and meta-analysis was not previously registered. 

## 5. Conclusions

The use of decision-making training based on the cognitive perspective is recommended for improving the decision-making of volleyball players. These programs can be focused on improving memory-related processes via image and video viewing, feedback, or video feedback and questioning, as these methods promote cognitive participation and reflection [40]. Such programs can also lead to improvements to visual and temporal parameters via simulation training or perceptual training. These programs aim to develop the anticipation and decision-making skills of the athlete by improving visual search strategies [43]. These findings will be useful for volleyball coaches as they emphasize the usefulness of such strategies when included as part of, or complementary to, regular active volleyball training for optimizing the decisional capacity of athletes. This type of training will allow players to select and process the most relevant stimuli in the environment and generate faster and more effective responses in different situations [50].

## Figures and Tables

**Figure 1 ijerph-17-03628-f001:**
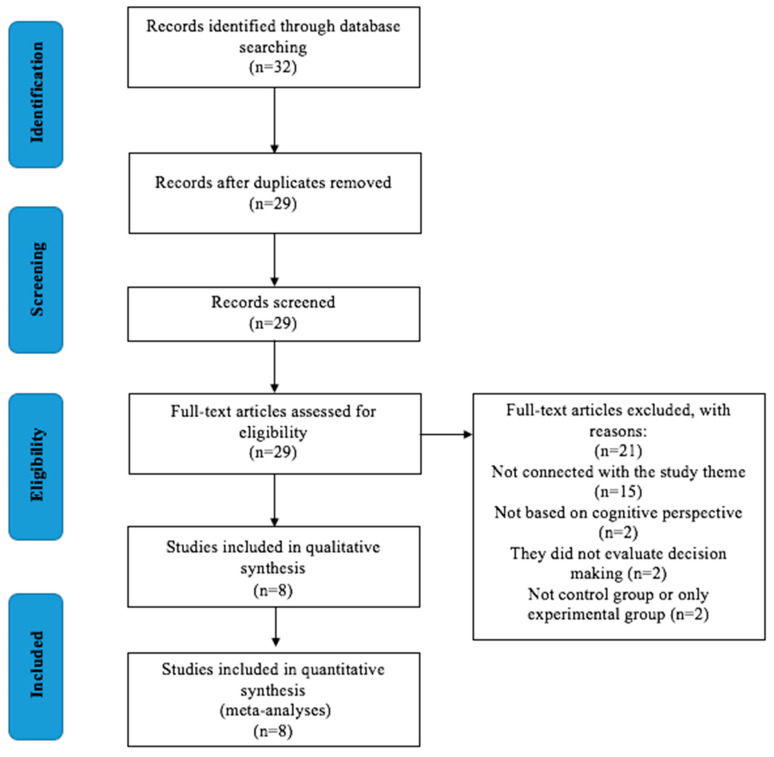
Summary of the search and study selection following PRISMA guidelines.

**Figure 2 ijerph-17-03628-f002:**
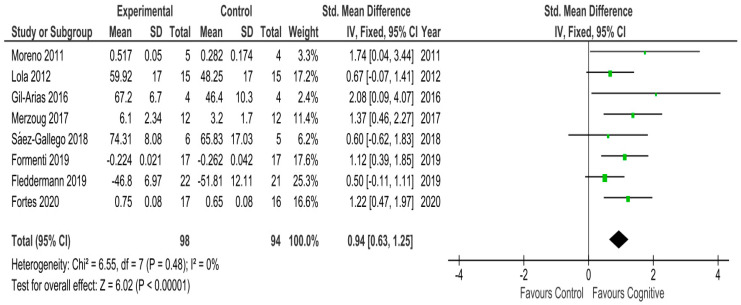
Meta-analysis of the effect of decision-making training programs/intervention, based on cognitive perspective, on decision making.

**Table 1 ijerph-17-03628-t001:** Characteristics of participants and study design.

Study	Participants	Age	Level of Play	Design	Country
Fleddermann et al., 2019 [54]	GE ^1^: 22 (2 men and 20 women) GC ^2^: 21 (5 men and 16 women)	Under-19	Players from 1st to 3rd division	Non-randomized controlled trial	Germany
Formenti et al., 2019 [55]	GE: 17 women GC: 17 women Only data from this group GEV ^3^ were used in the meta-analysis:17 women	Under-12	Regional League participants (minimum 4 years playing)	Randomized controlled trial	Italy
Fortes et al., 2020 [49]	GE: 17 men GC: 16 men	Under-17	Participants State Volleyball Championship	Randomized controlled trial	Brazil
Gil-Arias et al., 2016 [65]	GE: 4 women GC: 4 women	Under-16	Regional League Players	Non-randomized controlled trial	Spain
Lola et al., 2012 [66]	GI ^4:^15 women GS ^5:^15 women GC: 15 women Only data from this group GEX ^6^ were used in the meta-analysis:15 women	Under-12	Volleyball club players with 20 minimum workouts	Randomized controlled trial	Greece
Merzoug et al., 2017 [67]	GE: 12 men GC: 12 men	Under-17	Regional League Players	Non-randomized controlled trial	Algeria
Moreno et al., 2011 [40]	GE: 4 men GC: 4 men	Under-16	Regional League Players	Non-randomized controlled trial	Spain
Sáez-Gallego et al., 2018 [43]	GM ^7:^5 women GC: 5 women Only data from this group GV ^8^ were used in the meta-analysis: 6 women	Under-19	Regional League Players	Non-randomized controlled trial	Spain

^1^ Experimental Group, ^2^ Control Group, ^3^ Experimental Group Volleyball, ^4^ Implicit Group, ^5^ Sequential Group, ^6^ Explicit Group, ^7^ Mixed Group, ^8^ Video Group.

**Table 2 ijerph-17-03628-t002:** Characteristics of intervention, comparison group and outcome measure.

Study	Intervention	Comparison	Outcome	Duration of the Intervention
Fleddermann et al., 2019 [54]	Three-dimensional multi-object training (3D-MOT)	Regular active training	Processing speed	8 weeks 16 sessions 30 min/session
Formenti et al., 2019 [55]	Perceptual training through visual search strategies	Regular active training	Cognitive performance	8 weeks 30 min/session
Fortes et al., 2020 [49]	Imaging training program	Sports ad videos	Decision making in setting	8 weeks 24 sessions 10 min/session
Gil-Arias et al., 2016 [65]	Video-feedback and questioning program	Regular active training	Decision making in attack	11 weeks 60 min/session
Lola et al., 2012 [66]	Training through videos, execution demonstrations and instructions	Regular active training	Decision making in serve	4 weeks 12 sessions 70 min/session
Merzoug et al., 2017 [67]	Perceptual simulation training	Regular active training	Decision making effectiveness	Not reported
Moreno et al., 2011 [40]	Video-feedback and questioning program	Regular active training	Quality of decision making	13 matches 13 sessions
Sáez-Gallego et al., 2018 [43]	Perceptual training through video	Regular active training	Decision making in block	4 weeks 8 sessions 20 min/session

**Table 3 ijerph-17-03628-t003:** Risk of bias according to the evidence project risk of bias tool.

	1	2	3	4	5	6	7	8
Moreno 2011 [40]	Y	Y	Y	N	N	Y	Y	Y
Lola 2012 [66]	Y	Y	Y	Y	N	Y	Y	Y
Gil-Arias 2016 [65]	Y	Y	Y	N	N	Y	Y	Y
Merzoug 2017 [67]	Y	Y	Y	N	N	Y	Y	N
Saez-Gallego 2018 [43]	Y	Y	Y	N	N	Y	Y	N
Formenti 2019 [55]	Y	Y	Y	Y	N	Y	Y	Y
Fleddermann 2019 [54]	Y	Y	Y	N	N	Y	?	Y
Fortes 2020 [49]	Y	Y	Y	Y	N	Y	Y	Y

1. Cohort, 2. Control or comparison group, 3. Pre-post intervention data, 4. Random assignment of participants to the intervention, 5. Random selection of participants for assessment, 6. Follow-up rate of 80% or more 7. Comparison groups equivalent on sociodemographics, 8. Comparison groups equivalent at baseline on outcome measures.

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
