# Peer review of "The Effect of Decision Training, from a Cognitive Perspective, on Decision-Making in Volleyball: A Systematic Review and Meta-Analysis"

_ijerph, 2020, doi:10.3390/ijerph17103628_

Round 1
Reviewer 1 Report
Thank you for inviting me to review the study " The Effect of Decision Training, from a Cognitive Perspective, on Decision-Making in Volleyball: A Systematic Review and Meta-Analysis". The study provided interested results and information for research on cognitive processes. It could provide evidence for using decision training in sports. I have read carefully and found that this systematic review is very carefully created and developed. Although this study has scientific interest, some important aspects should be reviewed by the authors. I hope that my opinions will help shape your research article more precise and interesting. The followings are my comments:
Comments:
General comment: Please, review PRISMA Checklist (Moher et al, 2009).
- Introduction: The theorical background is clear.
- Objective: Provide an explicit statement of questions being addressed with reference to participants, interventions, comparisons, outcomes, and study design (PICOS).
- Methods: In reference to protocol and registration, please, indicate if a systematic review protocol exists, if and where it can be accessed (e.g., Web address), and, if available, provide registration information including registration number.
- Methods/inclusion criteria: Specify characteristics of the studies following PICOS, giving rationale.
- Methods/inclusion criteria: Move “To reduce selection bias, each study was independently reviewed by two of the authors (M.C.S. and C.F.-E.), who mutually determined whether or not they met basic inclusion criteria. If a consensous could not be reached on inclusion of a study, the matter was settled by consultation with a third author (M.P.M.A.).” to “study selection” section.
- Methods: Detail exclusion criteria
- Methods: Include “study selection” and “data collection process” sections
- Methods: Detail in the database that you pilot the search.
- Methods/ risk of bias in individual studies: The reference 56 is not appropriate, it is a general book about developing systematic reviews. Please, detail the tool that you have used for assessing the risk of bias. It is depending of the design of the individual studies.
- Methods/Table 1: Move Table 1 to results section.
- Methods: From line 159 to 216 move to result section.
- Results: Move 3.3 “study characteristics” to 3.2.
Author Response
First, we really appreciate the suggestions and comments made by the two reviewers. We truly believe that the quality of the manuscript has been improved after this revision.
All changes we made in the manuscript are in red.
Below, you can find the way we have addressed your suggestions and concerns:
- Revisor: “Objective: Provide an explicit statement of questions being addressed with reference to participants, interventions, comparisons, outcomes, and study design (PICOS)”.
Thank you for your comment. We have added the following objective including PICOS: “The current systematic review and meta-analysis aimed to review randomized and non-randomized controlled trials (S) to evaluate whether decision training from the cognitive approach (I) are more effective than usual training (C) to improve decision making (O) in volleyball players (P)” (Lines 118-121).
- Revisor: “Methods: In reference to protocol and registration, please, indicate if a systematic review protocol exists, if and where it can be accessed (e.g., Web address), and, if available, provide registration information including registration number”.
The protocol of this systematic review was not previously registered. In the “discussion”section, we have included as a limitation the lack of a previous registration (Lines 367-368).
- Revisor: “Methods/inclusion criteria: Specify characteristics of the studies following PICOS, giving rationale”.
Thank you for your comment. We realize this comment helps the reader. We have included the information in two tables. In Table 1, we include information about participants (P) and study design (S), while in Table 2 we include the characteristics of the intervention (I), comparison group (C) and the outcome measure (O)
- Revisor:“Methods/inclusion criteria: Move “To reduce selection bias, each study was independently reviewed by two of the authors (M.C.S. and C.F.-E.), who mutually determined whether or not they met basic inclusion criteria. If a consensous could not be reached on inclusion of a study, the matter was settled by consultation with a third author (M.P.M.A.).” to “study selection” section”.
Following your suggestions, we have moved that paragraph to the section “study selection and data collection”(Lines 153-156).
- Revisor: “Methods: Detail exclusion criteria”.
You were right that information was missing. We have added it and changed the title of the section to “inclusion and exclusion criteria”(Lines 132-134).
- Revisor: “Methods: Include “study selection” and “data collection process” sections”.
We have included in the “Methods”section the “Study Selection and Data Collection” (Lines 152-160).
- Revisor: “Methods/ risk of bias in individual studies: The reference 56 is not appropriate, it is a general book about developing systematic reviews. Please, detail the tool that you have used for assessing the risk of bias. It is depending of the design of the individual studies”.
You are completely right. We have modified the tool used to evaluate risk of bias. We have used the “Evidence Project” risk of bias tool, which can be used for both randomized and non-randomized controlled trials. We have added the proper reference of the journal article (Lines 147-151).
- Revisor: “Methods/Table 1: Move Table 1 to results section”.
Thank you, we have changed the tables in order to better report data following the PICOS approach. The two new tables are in the “results” section.
- Revisor: “Methods: From line 159 to 216 move to result section”.
We have moved it according to your suggestions (Lines 201-258).
- Revisor: “Results: Move 3.3 “study characteristics” to 3.2”.
We have fixed that, thank you.

Reviewer 2 Report
ABSTRACT:
30. "Training" duplicate
INTRODUCTION:
47-50. "Because of this, studies in sport have addressed decision-making from different perspective. Most have tried to understand and explain the decision-making process in sport to improve performance [6]. The two fundamental perspectives for the study of these decision-making processes are: the ecological perspective [25] and the cognitive perspective [26]". Please rewrite. there is a misunderstanding.
Add characteristics of volleyball in relation to cognitive demands and decision making to justify the objective of the study
METHODS
Table 1. In relation to the age and level of play... If all the studies are in young players, may be you should talk about the specific characteristics of young athletes in the introduction and also include it in the title. Did you find any study to match the inclusion criteria in adults or professional players?
159-216. It would be possible to present the decision-making training intervention details in a table? I will be easier to follow.
Exclude beach volleyball? It s not clear
RESULTS:
3.3 Study characteristics
Table 1 doesn´t show what is written in lines 253-255
Figure 3. Unclear. Difficult to read.
Which are the improvements after the training? Any specific situation during the game?
DISCUSSION:
295-304. Specify the situations in which their decisions are better. Specify the situations in which they reduce the time in reaction...
323-327. No simultaneous participation of the rival in the action to prevent it
REFERENCES:
49. Misquote (2020)
Should this paper have been included?
Wright, D. L., Pleasants, F., & Gomez-Meza, M. (1990). Use of advanced visual cue sources in volleyball. Journal of Sport and Exercise Psychology, 12(4), 406-414.
Author Response
First, we really appreciate the suggestions and comments made by the two reviewers. We truly believe that the quality of the manuscript has been improved after this revision.
All changes we made in the manuscript are in red.
Below, you can find the way we have addressed your suggestions and concerns:
- Revisor: “ABSTRACT: "Training" duplicate”.
Following your suggestion, that duplicate word has been removed.
- Revisor: “INTRODUCTION:Add characteristics of volleyball in relation to cognitive demands and decision making to justify the objective of the study”
We agree this is important. So, we have added a paragraph in the “introduction”section (lines 106-111) showing the decisional characteristics of volleyball.
- Revisor: “METHODS: Table 1. In relation to the age and level of play... If all the studies are in young players, may be you should talk about the specific characteristics of young athletes in the introduction and also include it in the title. Did you find any study to match the inclusion criteria in adults or professional players?”.
You are right, all the included studies involved young athletes. However, that was because we could not find any article with adult volleyball players that met the inclusion criteria. We did not include this in the title or the “introduction” section because it was not an inclusion criterion.
We agree it is relevant, so we have added a paragraph in the “discussion”section stating that there are not studies aimed to improve the decisional making process in adult volleyball players (lines 342-345).
- Revisor:“METHODS: 159-216. It would be possible to present the decision-making training intervention details in a table? I will be easier to follow”.
Following your suggestions, we have included a column in Table 2, summarizing the duration details of the intervention programs in order to make it easier to follow.
- Revisor:“METHODS: Exclude beach volleyball?It s not clear”.
Yes, beach volleyball was excluded but we agree it should have been stated in the manuscript. We have added it as a exclusion criterion (line 133).
- Revisor: “RESULTS: 3 Study characteristics. Table 1 doesn´t show what is written in lines 253-255”.
We agree with your comment. We have fixed that mistake.
- Revisor:“RESULTS: Figure 3. Unclear. Difficult to read”.
Figure 3 was made using the software RevMan 5.3, which is one of the most widely used and recommended tools to conduct meta-analysis. The figure includes the proper meta-analysis graph and a table with the data from experimental and control groups and also a calculation of SMD (95% CI). This figure is usually reported in scientific articles as we do in the current manuscript. We have also improved the resolution of the image.
- Revisor: “RESULTS: Which are the improvements after the training? Any specific situation during the game?”.
Following your comment, we have added in “results”the specific improvements that would increase the decision making in the different situations during the game. We have also added information in the “discussion”section (see below) (lines 286-288 and lines 357-359)
- Revisor: “DISCUSSION: 295-304. Specify the situations in which their decisions are better. Specify the situations in which they reduce the time in reaction”.
We have provided some examples of specific game situations in which the improvement in decision making and reaction time may lead to a better performance according to previous studies (lines 318-324).
- Revisor: “DISCUSSION: 323-327. No simultaneous participation of the rival in the action to prevent it”.
Although the two teams are separated by a net, there is a simultaneous participation due to each player adapt his/her behavior according to the changes and movements in the opponent team, which generates a large number of stimulus that must be perceived, selected and interpreted to perform the most adequate action.
- Revisor: “REFERENCES: Misquote (2020)”.
Thank you for pointing out that mistake. We have fixed it.
- Revisor: “REFERENCES: Should this paper have been included? Wright, D. L., Pleasants, F., & Gomez-Meza, M. (1990). Use of advanced visual cue sources in volleyball. Journal of Sport and Exercise Psychology, 12(4), 406-414”.
Following your suggestion, we have added a paragraph on the advantages of using this kind of techniques, citing the mentioned article (lines 357-359).
Round 2
Reviewer 1 Report
The authors has responded to comments appropriately.